# Continuing Education through the Campus Game: A Sustainable Gamification Project to Improve Doctors’ and Nurses’ Knowledge of Quality and Clinical Risk Management

**DOI:** 10.3390/healthcare11162236

**Published:** 2023-08-09

**Authors:** Claudio Pensieri, Anna De Benedictis, Francesco De Micco, Sabrina Saccoccia, Dhurata Ivziku, Marzia Lommi, Rossana Alloni

**Affiliations:** 1Department of Human Science, Libera Università Maria S.S. Assunta, Via Traspontina 21, 00193 Rome, Italy; 2Clinical Direction, Fondazione Policlinico Universitario Campus Bio-Medico, Via Alvaro del Portillo 200, 00128 Rome, Italy; a.debenedictis@policlinicocampus.it (A.D.B.); f.demicco@policlinicocampus.it (F.D.M.); s.saccoccia@policlinicocampus.it (S.S.); r.alloni@unicampus.it (R.A.); 3Department of Medicine and Surgery, Università Campus Bio-Medico di Roma, Via Alvaro del Portillo 21, 00128 Rome, Italy; 4Department of Healthcare Professions, Fondazione Policlinico Universitario Campus Bio-Medico, Via Alvaro del Portillo 200, 00128 Rome, Italy; d.ivziku@policlinicocampus.it; 5Department of Biomedicine and Prevention, University Tor Vergata, Via Cracovia 50, 00133 Rome, Italy; marzia.lommi@gmail.com

**Keywords:** adult education, healthcare, gamification, quality, risk management, medical education

## Abstract

The COVID-19 disease has dramatically changed lives worldwide, including education. This is a challenge for traditional learning. In fact, the European Higher Education Area poses the challenge of boosting the quality of teaching through active methodologies supported by digital pedagogy. Gamification is one of these tools and it has considerable attention in the healthcare literature. We aimed to create a game in the Campus Bio-Medico University Hospital Foundation in order to offer continuing education on Quality and Clinical Risk procedures to our staff. The 2021 “Campus Game” (178 players) introduced the “Badge Challenge” (Team Building, Procedures, and Security) and 73 questions. The leaderboard of every single match was posted in some of the hospital’s strategic areas and also published online on the company intranet to ensure engagement and competitiveness. Gamification has spontaneously promoted teamworking and a virtuous process of multiprofessional education. We found that, during the Campus Game, there was a 4.9% increase in access to the intranet page containing information on Quality and Patient Safety and an 8% increase in access to the Hospital Policies and Procedures. In the near future, we wish to expand this game, involving hospitals with similar types of activity and levels of attention to quality and safety issues, and also to enhance the network of partners and the principles of Q&S management itself.

## 1. Introduction

The COVID-19 disease has dramatically changed lives worldwide, including education [1]. Social distancing to control the spread of COVID-19 created a challenge for traditional learning [2]. We need to develop alternative methods of delivering education and continuing education (CE) in healthcare to optimize learning [3]. Gamification is a tool that has received considerable attention in the literature [4,5].

The European Higher Education Area poses the challenge of boosting the quality of teaching through active methodologies supported by digital pedagogy [6]. However, the reality is that, nowadays, the traditional model of master class teaching is still present and continues to be developed [7], so we aimed to involve hospital staff in a more efficient and funny way of learning.

This paper is a case history about a gamification experience used as an innovative method for continuing education on the quality and clinical risk management training at a University Hospital in Rome, Italy.

### Gamification

In order to define “Gamification”, a review on Pubmed was performed using “Gamification” as the main key term, including all articles from 2012 to January 2022 in all languages, and excluding articles about clinical application (a computer program or piece of software designed for a particular purpose that can be downloaded onto a mobile phone or other mobile device), healthcare mobile application, rehabilitation, and patient applications. In total, 929 articles were found and 113 articles for “Gamification + Review” were retrieved. The authors read the abstracts of the reviews, removing duplicates and off-topic papers. Only 7 articles were found about gamification in medical/adult education [2,8,9,10,11,12,13]. Gamification is a relatively new trend that focuses on applying game mechanics to non-gaming contexts [14] in order to engage audiences and inject a bit of fun into learning activities, as well as generate motivational and cognitive benefits [8]. Many fields, such as business, marketing, and e-learning, have already taken advantage of the potential of gamification. Likewise, the digital healthcare domain has also started to exploit this emerging trend. Gamification is receiving worldwide attention across many sectors as a powerful technique for promoting engagement and motivation [15,16,17,18,19,20]. It is necessary to distinguish between the different ways the game can be applied in contexts “outside” of the normal game:Serious Game: according to Stokes [21], serious games are designed to entertain players when they educate, train, or change their behavior. Serious games are developed for educational purposes and provide both realism and the entertainment facet of traditional games. This is a game with a purpose. It is a full-fledged game used for non-entertainment purposes. It has all the elements of a real game, and will look and feel like a real game, but has some defined purpose, outcome, or message that the creators wish to get across to you. It includes many educational games, as long as they have all the elements you would expect to see in a real game. Serious games have been also used to enhance learning in medical contexts [22,23].Alternative reality games (ARG): these connect the internet to the real world through numerous web tools (blogs, e-mails, or mini-websites). They include a mysterious story with clues that point to the real world (for example, monuments or hidden objects in certain locations). An ARG was used in the USA to launch the HALO 2 videogame, by the Nine Inch Nails band to launch Year Zero, and for “AI: Artificial Intelligence” and the “Batman The Dark Knight” films.Advergame: this is made specifically to communicate advertising messages, develop brand awareness, and generate website traffic. It is a free online or downloadable game. It is a pull advertising tool that guarantees a higher level of involvement and user exposure to the advertised brand than other advertising tools used on old and new media. It is generally broadcast through the advertiser’s website, on its social network pages, or is downloadable as an app for mobiles and other devices.Mobile applications (app): we can distinguish these between two types:○Medical apps (similar to medical devices) that monitor, control, or transform data that represent a patient’s physiological parameters, and therefore form an integral part of medical examinations (apps that measure blood pressure or perform eye examinations, help patients to manage chronic diseases, or calculate the correct dose of insulin for diabetics).○Health apps that work on a patient’s motivation for self care [24]. They can give access to clinical information, which supports the sharing of images of injuries, or they can comprise clinical diaries integrated with the healthcare provider. Alternatively, health apps can provide information on the interaction between drugs or create reminders for taking medication, influencing a patient’s lifestyle [25].Gamification: the most commonly accepted definition amongst those proposed is that of Sebastian Deterding, who stated that ‘‘Gamification is the use of game design elements in non-game contexts” [26]. Thus, rather than creating immersive, fully fledged games as in ‘‘serious games”, gamification is intended to affect users’ behavior and motivation by the means of experiences reminiscent of games [27]. Conceived in the digital media industry, gamification only began to be adopted on a large scale in the second half of 2010 [26]. In fact, its first documented use dates back to 2008, under the term ‘‘funware”, which was coined by Gabe Zichermann [20]. Gamification appeals to younger generations who are comfortable with smartphones, tablets, and other technology [28].

In our hospital, there are many employees born after 1997 and they are part of the so-called “Generation Z” (or Gen Z). This name refers to “digital natives” born in developed countries, who are currently the largest generational group of nurses, young doctors, and students of medicine [2]. Moreover, we have to keep in mind that they have never experienced life without the Internet and are constantly adapting to new technology [29]. Gen Z members have different motivations, learning styles, and skill sets compared to previous generations [30]. Gen Z learners frequently interact in the digital world and favor online learning [31]. They are anti-lecture and prefer to learn by solving real-world problems [29]. This could constitute a challenge if educators have not incorporated technology into their teaching style. Educators need to provide an array of innovative learning methods using various technological platforms to keep Gen Z learners engaged, and the use of digital gamification could be particularly beneficial to these digital natives to enhance their learning outcomes. Therefore, we decided to follow the gamification rules in our learning organization, using digital technology in order to offer to our staff a tool for Continuing Education on Quality and Clinical Risk procedures. In our Campus Game, we used large companies’ most-used methodologies (they used them to achieve the most varied business objectives: increase revenue, customer loyalty, and motivate the sales force, etc.):Points: assigning points to teams (7–12 employees) can motivate them to continue using a product or perform a certain action until they reach a more advanced level. These are generally used to subsequently obtain virtual or real goods.Badges: these are simple virtual emblems that demonstrate a certain ability of the user to perform something. They are used to make the user feel important and skilled and they strengthen their connection with the brand. They are one of the most-used gamification mechanics.Leaderboards: these are very useful in order to create competition between users. In this way, they are encouraged to use the service, thus spending more time within it, enabling them to compete with other people and obtain other new rewards in order to satisfy their final desire: to be at the top of the ranking.

Evidence has shown that a commitment-based, intrinsically motivated approach to change is more successful and sustained over time than a compliance-based (e.g., mandated) approach [32]. Gamification engages the user, provides a level of enjoyment, and incentivizes the learner to use critical thinking skills. However, the use of gamification (the application of game design and elements) as a clinical engagement strategy in health quality improvement initiatives is less explored [33].

## 2. Materials and Methods

Staff recruitment: in the FPUCBM hospital, we have 3 annual initiatives that encourage our staff to participate in continuous improvement processes:Quality Award: every year, the best 3 quality and safety projects from any department are awarded;Best monthly incident reporting: every month, the best incident reporting made by any one of the hospital’s staff is awarded;Campus Game: every year, we open the Campus Game to hospital staff (MDs and RNs, but also administrative staff, biologists, and pharmacologists, etc.), so everyone can play for free. The only enrollment criteria is to have a contract with our hospital.

In order to make the game sustainable, both economically and in terms of the use of consumable materials, we decided not to use paper and chose to use mobile phones. This led us to save costs in terms of sustainability, because: we did not use paper or company PCs and we did not use our classrooms that we usually use for traditional training (savings in electricity, heating, and time taken away from clinical care practice, etc.). We decided to use Google Forms for this game, because it is free and very versatile; moreover, our hospital uses the employee office suite, so data are secure within the organization. The enrollment criteria for the teams were: min. 3 and max. 9 people; we asked our staff to create multidisciplinary teams in order to increase corporate team building, even between different professions. The players received 3 multiple choice questions every Monday at 10.00 a.m. They had time to answer until 10.00 a.m. on Wednesday (48 h was necessary, taking into account the shifts of the health personnel). On Wednesday, after we closed the online questionnaire, we sent the correct answers to all the participants in order to ensure training, even for those who had not had the opportunity to answer.

The average responses of all the team members formed the final team’s score. Each match had 3 questions, from a minimum of 6 pt. to a maximum of 10 pt. per team, except the final match, which had 10 questions (instead of 3) for a total of 20 pts. We chose to give more points at the end in order to ensure the participation of teams that thought that, “mathematically”, they could not win (and therefore stop playing). The leaderboard of every single match was posted in some of the hospital’s strategic areas, near devices to stamp attendance (presence), near the café, near the self-service restaurant, and in the locker rooms. They were also published online on the company intranet to ensure engagement and competitiveness. Descriptive statistics of the means, frequencies, and percentages of the sample’s demographic characteristics and items were performed. We used the Microsoft Excel formula in order to do obtain these data.

### Novelties

Compared to the 2 previous editions, in 2021, we introduced three “BADGE CHALLENGE” (“Team Building”, “Procedures”, and “Safety”). Each badge granted the team 10 points. The tournament was divided into two rounds (like Italian football tournaments). The “first round” began in March 2021 and included 13 weeks + 2 badges, and it ended before the summer. The “second round” began after the summer in September 2021 and included 9 weeks + 1 badge, ending in November 2021. Thus, our staff answered 73 questions in 22 matches

Every question was created by specialists from different areas and were divided as follows (Table 1):

Some of these questions were:A patient’s family members report that they want to start the path toward a nursing home. What are you doing?A patient you have been treating for a long time sends you an email or a WhatsApp message with the following text: “Doctor, I have a problem, since I’ve been taking metformin, with the dosage you gave me, I feel dizzy, can I reduce it to half the dosage?”In our Polyclinic, who monitors the Point of Care Testing?You are making a handover communication. What method do you use?What are the areas that make up the CLABSI prevention bundle?What should the Data Controller do in the event of a personal data breach that may have had a significant impact on the personal data they process? Who can report an event (near-miss, sentinel event, and incident) on the company intranet?Where can you find how many people fell during the last quarter in our hospital?

Finally, we administered an assessment test that the participants filled out during the first week and last week.

We decided to use a 0 to 10 scale rating and the Likert Scale. We used the first one in order to obtain a judgment of quality (with 11 rating options, the 0-to-10 scale gave us a true average rating. According to our opinion, we think it is important to offer a true “average” option so respondents could indicate when something was neither exceptional nor poor). Than, we used the Likert scale in order to the evaluate players’ attitudes, opinions, and perceptions.

Those questions were not from a validated questionnaire, but we built them specifically for this Campus Game.

We used this in order to obtain data on “how much” the game helped people to acquire new information, theoretical knowledge, or improved their technical–professional skills.

## 3. Results

There were 178 players (118 females and 60 males) of this Campus Game. Females were almost double that of males, the reason for this probably being that, in our hospital, there are more female employees than males, but we did not investigate this aspect.

The roles of the players were: 154 healthcare workers (28 senior consultants/registrars, 60 residents, and 66 nurses), 17 allied health professionals (4 healthcare technicians, 4 biologists, 2 physical therapists, and 7 social health workers), 3 pharmacists, and 4 administrative support workers. The participants were divided into 21 multidisciplinary teams (Table 2, Figure 1)

The hospital intranet has web pages about quality and patient safety, hospital policies, and procedures, as well as a summary of the Joint Commission International accreditation standards manual for hospitals [34]. We compared the access to the intranet pages when the Campus Game was in progress versus when it was not in progress.

The Campus Game started in March 2021 and ended in November 2021, so we compared the percentage of access to the company intranet web pages before the game (January–February 2021) and during the game (March–November 2021). The data showed that, when the Campus Game was ongoing, there was an increase in access to the intranet web pages compared to when the Campus Game was not in progress: a 4.9% increase in access to the intranet page concerning Quality and Patient Safety compared to 0.3%; an 8% increase in access to the intranet page containing information on Hospital Policies and Procedures compared to 0.8%; and a 17.3% increase in access to the page with a summary of the JCI Accreditation Standards manual for hospitals compared to 0.3% (Table 3).

At the end of the game, the players answered the assessment form we described in the Methods paragraph.

To the question “Did you find Campus Game 2021 useful to increase your knowledge in the field of Quality and Patient Safety?”, the average score of the answers was 9.2 (rating scale: 0 = poor; 10 = excellent). The second question was: “Did you find interesting the Gamification system for this training activity?”, and the average score of the answers was 4.8 (rating scale: 0 = poor; 5 = excellent); The third question was: “Did you find fun use the Campus Game for your Continuing Education?”, and the average score of the answers was 4.8 (rating scale: 0 = poor; 5 = excellent) (Table 4).

## 4. Discussion

From the assessment questionnaires, it emerged that “gamification” (in its most intrinsic meaning) influenced and developed behavior centered on motivation and healthy competition, encouraging a sustainable “challenge” that was adaptable to the availability of time and resources of individual players.

The first three teams with the highest scores were awarded during the “Safety and Quality Day” in December 2021 in front of all the hospital’s staff. The prize consisted of money that they could spend on professional training. A team was also awarded for showing great commitment to finding answers. They had formally contested some answers, even with a literature search to support their reasons.

This was important for us, because it meant that we achieved the purpose of the game, which was not only “to have fun”, but “to train”, also through a spontaneous search for information. Above all, we wanted to create “engagement”. Thus, we can say that Gamification facilitated and promoted familiarity in acquiring information on company policies and procedures, thereby possibly improving the quality and safety of care. The involved players reported the usefulness of the Campus Game for the purposes of functional and motivational learning, with a view of improvement on the one hand and the motivation and performance of staff on the other.

In fact, as Alsawier [35] said: “players have greater motivation to learn, participate and collaborate” with interest and fun experienced when carrying out this training activity, which reflects what Colli [36] said: “you can also work in a fun way, enhancing the expressive component…”. On the other hand, knowledge and skills about company policies and procedures increased and this led to active participation and engagement with corporate objectives.

The data showed that one of the results of the Campus Game was the achievement of engagement with the use of digital tools. Teams were created with Generation Z (new digital) and people of the Net generation [37] and those from Generation X, who are more reluctant about digitization, overcoming the prejudice of traditional learning that has the limit of not falling within the sustainable development goals of the 2030 Agenda. Koulopoulos [38] said “it is recognized that Generation Z was born in a historical period in which natural resources are disappearing”. The economic crisis is not yet over and jobs are so scarce that Jeremy Finch [39], on the online business magazine Fast Company, said: “They only have the weight of saving the world and fixing our past mistakes on their small shoulders”. Generation Z has a great propensity for collaboration, sharing work aspects with a pragmatic sense of play with the goal of quick problem solving. At the same time, Gamification spontaneously promoted the start of a virtuous process of multi-professional education. In fact, a fundamental point is given by the social aspect that it entails: it is the “community” [40] that is created that makes the success of this type of approach possible.

Most of the Campus Game’s teams were composed of multi-professional teams in which doctors, nurses, technicians, administrative staff, and physiotherapists, etc. worked and “studied” together in order to achieve the same goal (that was to answer questions), by sharing their knowledge and skills. We want to underline the high number of participants despite the complexity and difficulties brought about due to the pandemic emergency. Gamification proved to be a precious resource and important educational tool, even in this extraordinary situation. Many professionals found in the “game” and healthy competition a diversion and tool for aggregation and sharing, which favored not only the growth of knowledge and skills, but also the teamworking ability and the ability to ask for help, in order to grow a critical mindset.

This is why we think that Gamification can be an “ecological educational resource”, a “game” that can favor the quality of personal and working life. After the final assessment questionnaire, some players suggested to us to introduce explanations of wrong answers, the performance of individual players in the teams, and shortening the time duration of the game.

Some limitations of our project are:It helped to generate interest, facilitate the location of professionally sensitive information and documentation, and stimulate and generate interest in learning, but this kind of Game did not guarantee learning (the use or putting into practice of the content). Thus, in the fourth edition, we started to verify in real life some generated changes, such as “where people and patients have to go in case of fire”, making players leave the hospital and take photos, or using the black box in order to test their adherence to hand hygiene rules, etc.We do not know if the people that accessed the company intranet studied, internalized, or put into practice the procedures and guidelines, etc., that they found. The only way we could verify this was when there was a dispute that a team made about the correct answers.. They studied the literature and, in some cases, did not agree with the answer. Then, after the explanations of the expert, they understood that they were wrong.

## 5. Conclusions

The permanent training of adult professionals is a constant challenge: everyone is convinced that it is necessary to update and resume notions and principles already known, but very few people do this in a constant and organized way. Even the imposition of specific levels of training has, so far, not achieved the desired results. However, the traditional paradigm of patient safety has been modified by the introduction of new technologies in healthcare [41], and there is need to emphasize the bio-social connections arising from the relationships between health conditions and socio-economic, political, and cultural determinants [42]. To cope with the needs to maintain a good level of knowledge and the application of quality and safety standards of care, our experience with the Campus Game was completely satisfactory, both in terms of participation and content acquisition. In the near future, we wish to expand this game, involving other hospitals with similar types of clinical activity and levels of attention to quality and safety issues, and also to share the principles of Q&S management.

## Figures and Tables

**Figure 1 healthcare-11-02236-f001:**
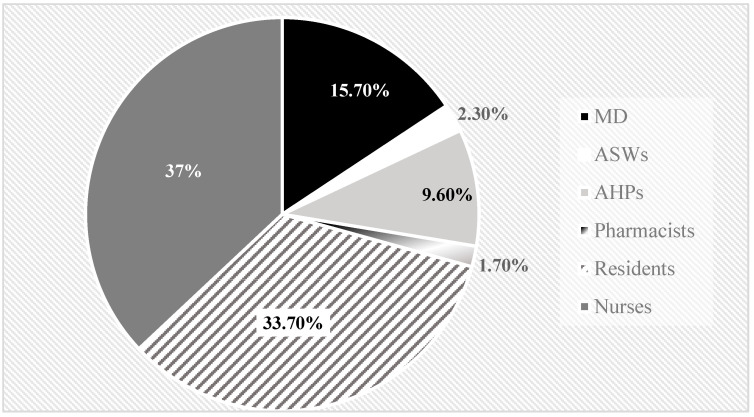
Campus Game players by role.

**Table 1 healthcare-11-02236-t001:** Number of questions per area.

AREAS	Tot.
Occupational Health and Safety	22
Infection Control	13
Privacy	10
Public Relations	5
Human Resources	2
Hospital Social Service	4
Quality and Clinical Risk Management	17
Tot	73

**Table 2 healthcare-11-02236-t002:** Campus Game players by role.

Teaching Hospital Participants by Role		n (%)
Healthcare Workers (HWs)	MD (Senior consultants/Registrars)	28 (15.7)
	Residents	60 (33.7)
	Nurses	66 (37)
Allied Health Professionals (AHPs)	Healthcare technicians	4 (2.3)
	Biologists	4 (2.3)
	Physical therapists	2 (1.1)
	Social health workers	7 (3.9)
Pharmacists		3 (1.7)
Administrative Support Workers (ASWs)		4 (2.3)
Total		178 (100)

**Table 3 healthcare-11-02236-t003:** Hospital intranet access.

	Quality and Patient Safety			Hospital Policies and Procedures			JCI Accreditation Standards for Hospitals (Summary)		
Campus Game in progress	11 March 2021	16 November 2021	Change (%)	11 March 2021	16 November 2021	Change (%)	11 March 2021	16 November 2021	Change (%)
	46.344	48.625	+2.281 (4.9)	45.236	48.864	+3.628 (8.0)	520	610	+90 (17.3)
Campus Game not in progress	14 January 2021	14 February 2021	Change (%)	14 January 2021	14 February 2021	Change (%)	14 January 2021	14 February 2021	Change (%)
	48.625	48.791	+166 (0.3)	48.864	49.280	+416 (0.8)	610	612	+2 (0.3)

**Table 4 healthcare-11-02236-t004:** Final Campus Game 2021 Assessment.

Did you find the Campus Game 2021 useful for increasing your knowledge in the field of Quality and Patient Safety?	Min. 0–Max. 10 Average
	9.2
Did you find the Gamification system interesting for this training activity?	Min. 0–Max. 5 Average
	4.8
Did you find the use of the Campus Game fun for your Continuing Education?	Min. 0–Max. 5 Average
	4.8

## Data Availability

Data sharing is not applicable to this article.

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
