# Peer review of "Continuing Education through the Campus Game: A Sustainable Gamification Project to Improve Doctors’ and Nurses’ Knowledge of Quality and Clinical Risk Management"

_healthcare, 2023, doi:10.3390/healthcare11162236_

Round 1

Reviewer 1 Report

This paper presents a case report on the implementation of gamification in the University Hospital's continuing education program for quality and clinical risk management training. The report aligns well with the journal's focus. However, I would like to suggest two minor revisions for the authors' consideration. Firstly, it would be beneficial to include details about staff recruitment and motivation for participation in the program. Secondly, in Table 3, the authors have computed the percentage change in Hospital intranet access but have not accounted for differences in time periods between the two groups, which may affect the validity of the results. The authors may wish to extend the time periods or provide an explanation for this discrepancy.

Minor editing of English language required.

Author Response

I would like to suggest two minor revisions for the authors' consideration. Firstly, it would be beneficial to include details about staff recruitment and motivation for participation in the program.

Thank you.

We have made this article changes:

Staff recruitment: in the FPUCBM hospital we have 3 annual initiatives that encourage our staff to participate in continuous improvement processes:

  1. Quality Award: every year the best 3 quality & safety projects from any depart-ment are awarded;
  2. Best monthly Incident Reporting: every month the best incident reporting made from anyone of hospital’s staff is awarded;
  3. Campus Game: every year we open the Campus Game to Hospital Staff (MD, RN, but also administrative, biologist, pharmacologist, etc. etc.) so everyone can play for free. The only enrollment criteria is to have a contract with our hospital.

 Secondly, in Table 3, the authors have computed the percentage change in Hospital intranet access but have not accounted for differences in time periods between the two groups, which may affect the validity of the results. The authors may wish to extend the time periods or provide an explanation for this discrepancy.

Thank you.

We have made this text changes:

Campus Game started in March 2021 and ended in November 2021, so we compared the percentage of access to company intranet web pages before the game (Janu-ary-February 2021) and during the game (March-November 2021).

Reviewer 2 Report

Dear authors,

I read with interest your study and I found it very original. As a supporter of game theories, I encourage every trial of implementing game characteristics in real world scenarios and appreciated your study.

However, I would like some extra information in order to proceed to a final judgment of your work (some are rather a suggestion to improve readability): 

1) line 55 --> not clear the meaning of app. (please avoid both abbreviation such as this one, and the contracted form like "didn't")

2)line 111-112 --> since not only Google Forms is free and versatile (?) , this justification sounds more an advertisement. I suggest to report this methodological choice either with technical details (maybe this aspects does affect responsiveness ?) or without this emphasis.

3)line 118 --> ALL is reported in capital letters. There is no need of it.

4)lines 120 and following --> can the authors provide the criteria for enrolment of participants? Did every worker have the same probability of participation?

5)in the material and methods section the data analysis plan is missing. Also, i think lines 165-172 sound more methodological. I suggest to report the questions and the evaluation metrics in the materials/methods and just leaving the results in the results section. Moreover, may the authors provide a reason for one question ranging from 0 to 10 and the others from 0 to 5? This is quite odd.

6)the quite significant imbalance in Female to Male ratio (118 to 60) is interesting but not commented. Is this due to the fact that the facility has more women than men employed or is there another reason behind this result?

7)the title declares an improvement in the knowledge of Quality and Clinical Risk Management, but the reader has limited capability to assess this feature, since no example question have been provided (there is also no supplementary material). Since only 73 question have been asked, my suggestion is to collect and report them all in a single supplementary material file, divided by topic. Alternatively, I suggest to report a sample of them in a table in the main text. 

8) references: some are rather cryptical, i.e. 23, 25, 28. I think that if they are concerning a book, an ISSN or ISBN code shoud be reported. Also, in line 210 the caption seems more to be a reference.

Kind Regards, 

can the authors provide a justification about 

Dear authors, 

I have found the English language pretty good. Some typos to be fixed, and I suggest not to use the contracted forms ("didn't", "don't",etc.).

Kind Regards,

Author Response

I read with interest your study and I found it very original. As a supporter of game theories, I encourage every trial of implementing game characteristics in real world scenarios and appreciated your study.

However, I would like some extra information in order to proceed to a final judgment of your work (some are rather a suggestion to improve readability): 

  • line 55 --> not clear the meaning of app.

Thank you, we have made this article's changes:

"About clinical application (a computer program or piece of software designed for a particular purpose that can be downloaded onto a mobile phone or other mobile device), healthcare mobile application, rehabilitation and patient’s applications".

(please avoid both abbreviation such as this one, and the contracted form like "didn't")

Ok, I have done it.

2) line 111-112 --> since not only Google Forms is free and versatile (?) , this justification sounds more an advertisement. I suggest to report this methodological choice either with technical details (maybe this aspects does affect responsiveness ?) or without this emphasis.

Thank you, we have made this article's changes:

"We decided to use Google Forms for this game, because it is free and very versatile, moreover our hospital uses the employee office suite, so the data is secure within the organization".

3)line 118 --> ALL is reported in capital letters. There is no need of it.

Right. Done. Thank you.

4)lines 120 and following --> can the authors provide the criteria for enrolment of participants? Did every worker have the same probability of participation?

Thank you, we made this article's changes:

  1. Campus Game: every year we open the Campus Game to Hospital Staff (MD, RN, but also administrative, biologist, pharmacologist, etc. etc.) so everyone can play for free. The only enrollment criteria is to have a contract with our hospital.

5)in the material and methods section the data analysis plan is missing.

Thank you, we made this article's changes:

"Descriptive statistics of means, frequencies and percentages for the sample’s demographic characteristics and items were performed".

Also, i think lines 165-172 sound more methodological.

I’m sorry, but I have not understood this suggestion. In my PC the paper lines are different... 

Please see the attached file. If you can tell me the phrase we have to change I'll do it.

I suggest to report the questions and the evaluation metrics in the materials/methods and just leaving the results in the results section.

Ok, I have done the change requested.

Moreover, may the authors provide a reason for one question ranging from 0 to 10 and the others from 0 to 5? This is quite odd.

Thank you, we made this article's changes:

"We decided to use a 0 to 10 scale rating and Likert Scale. We used the first one in order to have a judgment of quality (with 11 rating options, the 0-to-10 scale gives us a true average rating. It's important to offer a true “average” option so respondents can indicate when something was neither exceptional nor poor).  Than we used the Likert scale in order to evaluate players attitudes, opinions and perceptions".

6) the quite significant imbalance in Female to Male ratio (118 to 60) is interesting but not commented. Is this due to the fact that the facility has more women than men employed or is there another reason behind this result?

Thank you, we made this article's changes:

"Players of this Campus Game were 178, players (118 females; 60 males), females were quite the double of males, probably the reason is that in our hospital there are more female employees than males, but we have not investigated this aspect".

7)the title declares an improvement in the knowledge of Quality and Clinical Risk Management, but the reader has limited capability to assess this feature, since no example question have been provided (there is also no supplementary material). Since only 73 question have been asked, my suggestion is to collect and report them all in a single supplementary material file, divided by topic. Alternatively, I suggest to report a sample of them in a table in the main text. 

Thank you, we made this article's changes:

Some of this question were:

  1. A patient's family members report that they want to start the path in a nursing home. What are you doing?
  2. A patient you have been treating for a long time sends you an email or a whatsapp message with the following text: "Doctor, I have a problem, since I've been taking metformin, with the dosage you gave me, I feel dizzy, can I reduce it to half the dosage?"
  3. In our Polyclinic who monitors the Point of Care Testing?
  4. You are making the handover communication. What method do you use?
  5. What are the areas that make up the CLABSI prevention bundle?
  6. What should the Data Controller do in the event of a personal data breach that may have had a significant impact on the personal data he processes?Who can report an event (near-miss, sentinel event, incident) on the company intranet?
  7. Where can you find how many people fell during the last quarter in our hospital?

8) references: some are rather cryptical, i.e. 23, 25, 28. I think that if they are concerning a book, an ISSN or ISBN code shoud be reported.

Thank you, we made this article's changes:

18: Zichermann G., Linder J. The Gamification Revolution: How Leaders Leverage Game Mechanics to Crush the Competition. East Windsor, NJ: McGraw Hill Education, 2013. ISBN 9780071808316

23: Seemiller C., Grace M. Generation Z Goes to College. Wiley, San Francisco, CA, 2016, ISBN 1119143454

25: Joint Commission International, Accreditation Standards for Hospitals, 7th Edition, 1 January 2021. ISBN 9781635851489

27: Colli F., Meneghetti C., Viola F. Giocarsi: gaming e gamification in contesti professionali. Hogrefe, Firenze, 2021. ISBN 8898542658

28: Pensieri C., Game Therapy, Universitalia, Rome, 2013. ISBN 8865075236.

29: Koulopoulos T., Keldsen D. The Gen Z Effect: The Six Forces Shaping the Future of Business. Routledge, London, 2014. ISBN 9781629560311

Also, in line 210 the caption seems more to be a reference.

I’m sorry, but I have not understood this suggestion. In my PC the paper lines are different... 

Please see the attached file. If you can tell me the phrase we have to change I'll do it.

Reviewer 3 Report

Dear author, please find the following feedback

1.       ine 41-42 – need reference on ‘gamification is a tool that received considerable attention’

2.       Line 55-57- this detail on the systematic review was not required (screening)

3.       Authors use the systematic review to define gamification. The citation to the review was not provided (rather authors have cited individual studies included in the review). In addition, what was the conclusion of the review.

4.       The authors have jumped the concepts in a hurry. Page 2 paragraph 1 – started with definition of gamification but rapidly jumped into serious game and then provided definition of gamification. Needs to be avoided

5.       Authors have claimed that generation X have never experienced life without internet (lines 78-80). This is potentially fulfilled by those who were born and raised in the developed countries. We need to remember that more than 50% of people are from developing countries who may not even, today, have access to internet and other services.

6.       Authors have not defined what a match was. How many matches were played?

7.       Authors need to explain why they used different Likert scales in the questionnaire. It makes readers confused.

8.       What was the prize amount? Needs to be declared? Authors need to justify if the prize money was an incentive to participate in the research.

9.       What about the ethics, ethical approval, ethical considerations, and consent procedures?

Author Response

  1. Line 41-42 – need reference on ‘gamification is a tool that received considerable attention’

Lopes R.P, (2014) Gamification as a learning tool. International Journal of Developmental and Educational Psychology INFAD Revista de Psicología, 1(2): 565-574565. DOI: http://dx.doi.org/10.17060/ijodaep.2014.n1.v2.473

Koivisto J., Hamari J., (2019) The rise of motivational information systems: A review of gamification research, International Journal of Information Management, 45: 191-210, ISSN 0268-4012, https://doi.org/10.1016/j.ijinfomgt.2018.10.013.

  1. Line 55-57- this detail on the systematic review was not required (screening).

Ok, but we think it's useful to understand why “gamification” is becoming more and more important nowadays. But if you think those data are not important we can delete them.

  1. Authors use the systematic review to define gamification. The citation to the review was not provided (rather authors have cited individual studies included in the review). In addition, what was the conclusion of the review.

We cited this systematic review:

Min A., Min H., Kimm S. Effectiveness of serious games in nurse education: A systematic review. Nurse Educ Today. 2022. 108:105178. doi: 10.1016/j.nedt.2021.105178. Epub 2021 Oct 22. PMID: 34717098

Sardi L., Idri A., Fernandez-Aleman J.L. A systematic review of gamification in e-Health. Journal of Biomedical Informatics. 2017;71: 31–48.

Van Gaalen A.E.J.; Brouwer J., Schönrock‑Adema J., Bouwkamp‑Timmer T., Jaarsma A.D.C., Georgiadis J.R. Gamification of health professions education: a systematic review. Advances in Health Sciences Education, 2020, 26:683–711. https://doi.org/10.1007/s10459-020-10000-3

We need to cite some more systematic review? Or we can only cite them in better way in the paper?

  1. The authors have jumped the concepts in a hurry. Page 2 paragraph 1 – started with definition of gamification but rapidly jumped into serious game and then provided definition of gamification. Needs to be avoided

We made this article text's change:

"In order to define “Gamification” a review on Pubmed has been performed using “Gamification” as main key term, including all articles from 2012 to January 2022 in all languages and excluding articles about clinical application (a computer program or piece of software designed for a particular purpose that can be downloaded onto a mobile phone or other mobile device), healthcare mobile application, rehabilitation and patient’s applications. 929 articles were found and 113 articles for “Gamification + Review” were retrived. Authors read the abstracts of the Reviews removing duplicates and off-topic papers. Only 7 articles were found about gamification in the medical/adult education [2, 6- 11]. Gamification is a relatively new trend that focuses on applying game mechanics to non-gaming contexts [12] in order to engage audiences and inject a bit of fun into learning activities as well as to generate motivational and cognitive benefits [6]. Many fields such as Business, Marketing and e-Learning have already taken advantage of the potential of gamification. Likewise, the digital healthcare domain has also started to exploit this emerging trend. Gamification is receiving worldwide attention across many sectors as a powerful technique to promote engagement and motivation [13- 18].  It is necessary to distinguish between the different ways the game can be applied in contexts "outside" of the normal game:

  • Serious Game: according to Stokes [19], serious games are designed to entertain players when they educate, train or change their behavior. Serious games are developed for educational purposes and provide both realism and the entertainment facet of traditional games. It is a game with a purpose. It is a full-fledged game used for a non-entertainment purpose. It has all the elements of a real game, will look and feel like a real game, but has some defined purpose, outcome or message that the creators wish to get across to you. It includes many educational games, as long as they have all the elements we would expect to see in a real game. Serious games have been also used to enhance learning in medical contexts [35-36].
  • Alternative reality games (ARG): it connect the internet to the real world through numerous web tools (blogs, e-mails, mini-websites). A mysterious story with clues that points to the real world (for example, monuments or hidden objects in certain locations). ARG was used in the USA to launch the HALO 2 videogame, by the Nine Inch Nails band to launch Year Zero and for “AI: Artificial Intelligence” and “Batman The Dark Knight” films.
  • Advergame: it is made specifically to communicate advertising messages, develop brand awareness and generate website’s traffic. It is a free online or downloadable game. It is a pull advertising tool that guarantees a higher level of involvement and user exposure to the advertised brand than other advertising tools used on old and new media. It is generally broadcast through the advertiser's website, on its social network pages or is downloadable as an app for mobiles and other devices.
  • Mobile applications (App): we can distinguish between two types:
    • Medical apps (similar to Medical Devices) that monitor, control or transform data that represent the patient's physiological parameters, and therefore form an integral part of a medical examination (Apps that measure blood pressure or perform eye examinations, help the patient to manage chronic diseases, or calculate the correct dose of insulin for diabetics).
    • Health apps that work on the patient's motivation to self-care [37]. They can give access to clinical information, which supports the sharing of images of injuries, or they can comprise of clinical diaries integrated with the healthcare provider. Alternatively, health apps can provide information on the interaction between drugs or create reminders for taking medication and to influence the patient's lifestyle [38].
  • Gamification: the most commonly accepted definition amongst those proposed was that of Sebastian Deterding who stated that ‘‘Gamification is the use of game design elements in non-game contexts” [20]. So, rather than creating immersive, full-fledged games as in ‘‘serious games”, gamification is intended to affect the users’ behavior and motivation by means of experiences reminiscent of games [21]. Conceived in the digital media industry, gamification only began to be adopted on a large scale in the second half of 2010 [20]. In fact, the first documented use dates back to 2008, under the term ‘‘funware”, which was coined by Gabe Zichermann [18]. Gamification appeals to the younger generations who are comfortable with smartphones, tablets, and other technology [39].

In our hospital there are many employees born after 1997 and they are part of the so-called “Generation Z” (or Gen Z). This name is referred to “digital natives”, born in developed countries, that are currently the largest generational group of nurses, young doctors and students of Medicine [2]. Moreover we have to keep in mind that they have never experienced life without the Internet and are constantly adapting to new technology [22]. Gen Z members have different motivations, learning styles, and skill sets compared with previous generations [23]. Gen Z learners frequently interact in the digital world and favor online learning [24]. They are anti-lecture and prefer to learn by solving real-world problems [22]. This could constitute a challenge if educators have not incorporated technology into their teaching style. Educators need to provide an array of innovative learning methods using various technological platforms to keep Gen Z learners engaged, and the use of digital gamification could be particularly beneficial to these digital natives to enhance learning outcomes. So we decided to follow the gamification rules in our learning organization using digital technology in order to offer to our staff a tool for Continuing Education on Quality and Clinical Risk procedures. In our Campus Game we have used the large companies most used methodologies (they used them to achieve the most varied business objectives: increase revenue, customer loyalty, motivate the sales force, etc.):

  • Points: assigning points to Teams (7-12 employees) can motivate them to continue using the product or to perform a certain action until they reach a more advanced level. They are generally used to subsequently obtain virtual or real goods.
  • Badges: they are simple virtual emblems that demonstrate a certain ability of the user to do something. Decisive to make the user feel important and skilled, they strengthen their connection with the brand. They are part of the most used gamification mechanics.
  • Leaderboards: they are very useful in order to create competition between users. In this way they are encouraged to use the service, thus spending more time inside it, to be able to compete with other people and to obtain other new rewards in order to satisfy their final desire: to be at the top of the ranking.

Evidence has shown that a commitment-based, intrinsically motivated approach to change is more successful and sustained over time than a compliance-based (e.g. mandated) approach [40]. Gamification engages the user, provides a level of enjoyment, and incentivises the learner to use critical thinking skills. But the use of gamification (application of game design and elements) as a clinical engagement strategy in health quality improvement initiatives is less explored [41].

  1. Authors have claimed that generation X have never experienced life without internet (lines 78-80). This is potentially fulfilled by those who were born and raised in the developed countries. We need to remember that more than 50% of people are from developing countries who may not even, today, have access to internet and other services.

Thank you, you are right! We made this article's text change:

In our hospital there are many employees born after 1997 and they are part of the so-called “Generation Z” (or Gen Z). This name is referred to “digital natives,” born in developed countries, that are currently the largest generational……

  1. Authors have not defined what a match was. How many matches were played?

Thank you. We made this article's change:

So, our staff answered 73 questions in 22 matches.

  1. Authors need to explain why they used different Likert scales in the questionnaire. It makes readers confused.

Thank you. We made this article's change:

"We decided to use a 0 to 10 scale rating and Likert Scale. We used the first one in order to have a judgment of quality (with 11 rating options, the 0-to-10 scale gives us a true average rating. It's important to offer a true “average” option so respondents can indicate when something was neither exceptional nor poor).  Than we used the Likert scale in order to evaluate players attitudes, opinions and perceptions".

  1. What was the prize amount? Needs to be declared? Authors need to justify if the prize money was an incentive to participate in the research.

We do not want to make public the economic prize, we are talking about 10.000 Euros divided into: Quality Award; Best monthly Incident Reporting and Campus Game. Players do not know how much they will win. Every year we decide how to divide this 10.000 Euros for every award…

  1. What about the ethics, ethical approval, ethical considerations, and consent procedures?

We did not ask those documents to our Ethical Committee because it was not a research, but only a fun way to learn something… The public data that we uploaded in the company intranet was always anonymous and aggregated.

Reviewer 4 Report

I believe that this type of training initiatives are great for stimulating learning, they help to generate interest, facilitate the location of professionally sensitive information-documentation, stimulate and can generate interest in learning, but do not in themselves guarantee learning, the use or putting into practice of the contents. This would be the first limitation, not mentioned by the authors, that of not being able to contrast or verify the change generated, if the search for the game has been translated into internalization and putting into practice of the knowledge. We cannot know if it has served to acquire knowledge; ... it speaks of locating it (searching for information, contracting), not of studying it, internalizing it or putting it into practice.
These are positive experiences in the training strategy as a whole, but insufficient on their own.
The second limitation I find is the target population (generation z, as the authors allude to), forgetting that many other age groups with different customs, socio-educational practices and level of digital use are present in the hospital center. Have we thought about how to involve them, and if not, what recreational-training alternatives have been considered?
The question responds to the fact that while we can stimulate the participation of certain groups with gamification, we can generate a demotivating effect on those who do not feel (can not, have misgivings) involved in these practices.
On the other hand, I understand that the content of the experience presented is not scientific enough to be presented as a research article, but as a congress communication, since it is a real experience of teaching innovation, so I do not recommend its publication as an article, but in other formats of scientific academic type

Author Response

I believe that this type of training initiatives are great for stimulating learning, they help to generate interest, facilitate the location of professionally sensitive information-documentation, stimulate and can generate interest in learning, but do not in themselves guarantee learning, the use or putting into practice of the contents. This would be the first limitation, not mentioned by the authors, that of not being able to contrast or verify the change generated, if the search for the game has been translated into internalization and putting into practice of the knowledge. We cannot know if it has served to acquire knowledge; ...

Thank you! We made this article's change:

Some limitation of our project are:

  1. It helped to generate interest, facilitate the location of professionally sensitive in-formation-documentation, stimulate and generated interest in learning, but this kind of Game did not guarantee learning (the use or putting into practice of the content). We, in the 4th edition, started to verify on the floor some generated changes, like “where people and patients have to go in case of fire” we make them go out from hospital and we took photos, or we used the black box in order to test their adherence to hand hygiene rules, etc. etc.

it speaks of locating it (searching for information, contracting), not of studying it, internalizing it or putting it into practice.

We made this article's change:

  1. We do not know if people that accessed the company intranet had studied, internalized or put into practice procedures, guidelines, etc. that they found. The only way we had to do this was about the dispute that some team made about the correct answer to some questions. They studied the literature and in some cases did not agree with answer. Then, after the explanations of the expert they understood that they were wrong.

These are positive experiences in the training strategy as a whole, but insufficient on their own.
The second limitation I find is the target population (generation z, as the authors allude to), forgetting that many other age groups with different customs, socio-educational practices and level of digital use are present in the hospital center. Have we thought about how to involve them, and if not, what recreational-training alternatives have been considered?

Yes, but we did not mentioned it, for them we continued to have lectures and meetings in the wards.

The question responds to the fact that while we can stimulate the participation of certain groups with gamification, we can generate a demotivating effect on those who do not feel (can not, have misgivings) involved in these practices.

Thank you for this hint. We have not explored this aspect. Thank you for the advice, in the next campus game we will also do an assessment questionnaire open to those who "did not" participate to evaluate this aspect. Thanks so much for this great tip!

On the other hand, I understand that the content of the experience presented is not scientific enough to be presented as a research article, but as a congress communication, since it is a real experience of teaching innovation, so I do not recommend its publication as an article, but in other formats of scientific academic type.

Thank you for this hint, we presented the second Campus Game at the International Forum on Quality and Safety in Healthcare 2-6 November 2020. Copenhagen. BMJ & Institute for Healthcare Improvement.

But we would like to write it as an article in order to allow other hospital to know that this could be a new way to train people in the Quality and Risk Management. This year we are trying to assess it on the floor, to monitor the results. But we don’t have those data for the 2021 Campus Game.

Pensieri C, De Benedictis A, Nobile L, Iori T, Saccoccia S, Alloni R. (2020) The “2nd CAMPUS GAME” An effective educational Risk & Quality gamification project of the University Hospital Campus Bio-Medico. Poster presented at International Forum on Quality and Safety in Healthcare 2-6 November 2020. Copenhagen. BMJ & Institute for Healthcare Improvement. DOI: 10.13140/RG.2.2.13263.51366

Round 2

Reviewer 2 Report

Dear authors, 

please refer to the attached document.

Kind regards,

Dear authors, 

the english is now fine.

Kind Regards,

Author Response

Dear authors,

I read the updated manuscript. Here are my comments:

1) “Descriptive statistics of means, frequencies and percentages for the sample’s demographic characteristics and items were performed.” Lines 189-191.

The software used to analyze data is still missing. Did you use a specific software or just reported the built-in Google results?

I changed in this way:

Descriptive statistics of means, frequencies and percentages for the sample’s demographic characteristics and items were performed. We used Microsoft Excel formula in order to do obtain those data.

Is it ok now?

2) I’m sorry, but I have not understood this suggestion. In my PC the paper lines 165-172 are this: In the updated version of the manuscript the lines I was referring to are now 261-267, starting from “To the question "Did you find Campus …” to “0 = poor; 5 = excellent).”The questions formulated are the measurement tools that allow the researcher to assess the perception of the participant. So, I would have expected them to be put in the methods section,

Thank you, I moved this description in the Methods section.

along with a comment about how they were constructed (e.g. are they validated in literature? Do they derive from a validated questionnaire?).

Those questions were not from a validated questionnaire but we built them specifically for this Campus Game.

I suggest reporting the averages for every question in the result section, maybe in a table format.

Done. Thank you!

Table 4. Final Campus Game 2021 Assessment

Did you find Campus Game 2021 useful to increase your knowledge in the field of Quality and Patient Safety?

Min. 0 – Max. 10

Average

9.2

Did you find interesting the Gamification system for this training activity?

Min. 0 – Max. 5

Average

4,8

Did you find fun use the Campus Game for your Continuing Education?

Min. 0 – Max. 5

Average

4,8

3) It's important to offer a true “average” … line 258

Is it author’s opinion or is it believed to be important by scientific community? In the first case, please add “according to our opinion”. In the second case, a reference is needed (and it is a more robust assertion). Also, remember to avoid contracted form.

Yes!! Thank You.

I modified like this: “According to our opinion we think it is important…”

4) ”I’m sorry, but I have not understood this suggestion. In my PC the paper line 210 is this:” now line 305:

I’m not sure,but if it is a reference it should be put in the reference section, along with the other sources of information.

  1. www.24consulting.it/gamification-cose-e-a-cosa-serve accessed on 26/08/2022.

5) line 42 with the [33-34] reference: I suggest to reorganize the reference numbering so that the sequence is straight. Usually, the reference manger of the software used to compile the manuscript does this automatically, but something went wrong. Please check.

I tried.

6) line 282: “…thereby improving the quality and safety of care.”. This is a strong assertion for a case report, and no justification nor reference is reported to explain the link between acquiring information and improving the quality of care. I would reformulate as this: “… thereby possibly improving the quality and safety of care”.

Yes!!! Thank you for your suggestion! I used it:

procedures, thereby possibly improving the quality and safety of care improving the quality and safety of care. Involved

Also, since there is no inference analysis (no CI, no hypothesis testing), the authors should be cautious in generalizing the results.

Reviewer 4 Report

Theoretical and methodological clarifications have been introduced, giving more substance and solvency to the work presented. The obvious limitations that have been suggested are also included, but I insist that the specific experience is ideal content for a congress or teaching innovation experience, but not for an academic article, as it does not provide theoretical or systematic review results, nor qualitative or quantitative results that can be extrapolated to other realities. It does not advance the knowledge of gamification in clinical decision making. That is why I insist that it is not publishable as an article, but as a proceeding.

Author Response

Ok thank you.

We want to submit the Campus Game 2022 to the next Quality And Safety Conference in London 2024.

This article is about the 2021 Campus Game, we could use this article in order to make a better Poster in London...

Thank you for your suggestion.